# Key Factors and AI-Based Risk Prediction of Malnutrition in Hospitalized Older Women

**DOI:** 10.3390/geriatrics7050105

**Published:** 2022-09-26

**Authors:** Nekane Larburu, Garazi Artola, Jon Kerexeta, Maria Caballero, Borja Ollo, Catherine M. Lando

**Affiliations:** 1Vicomtech Foundation, Basque Research and Technology Alliance, 20009 San Sebastián, Spain; 2eHealth Group, Biodonostia Health Research Institute, 20009 San Sebastián, Spain; 3Asunción Klinika, 20400 Tolosa, Spain

**Keywords:** malnutrition, older adults, hospitalized patients, predictive model

## Abstract

The numerous consequences caused by malnutrition in hospitalized patients can worsen their quality of life. The aim of this study was to evaluate the prevalence of malnutrition on the elderly population, especially focusing on women, identify key factors and develop a malnutrition risk predictive model. The study group consisted of 493 older women admitted to the Asunción Klinika Hospital in the Basque Region (Spain). For this purpose, demographic, clinical, laboratory, and admission information was gathered. Correlations and multivariate analyses and the MNA-SF screening test-based risk of malnutrition were performed. Additionally, different predictive models designed using this information were compared. The estimated frequency of malnutrition among this population in the Basque Region (Spain) is 13.8%, while 41.8% is considered at risk of malnutrition, which is increased in women, with up to 16.4% with malnutrition and 47.5% at risk of malnutrition. Sixteen variables were used to develop a predictive model obtaining Area Under the Curve (AUC) values of 0.76. Elderly women assisted at home and with high scores of dependency were identified as a risk group, as well as patients admitted in internal medicine units, and in admissions with high severity.

## 1. Introduction

Several factors may contribute to an older individual suffering from nutrient deficiencies and other imbalances. For example, sensory disturbances, functional disabilities, and social isolation that typically accompany aging all increase the likelihood that a person will develop unhealthy eating habits. What is more, the mere fact of being female is one of the risk factors for developing these habits [1,2,3,4,5], which may lead to nutrient deficiencies or other imbalances that ultimately result in physical manifestations such as altered body composition and body cell mass. These physical changes cause (i) diminished physical and mental function, (ii) alterations in the immune system, (iii) worsening of the underlying disease, (iv) longer hospital stays and readmissions, and (v) a lower quality of life (QoL) [6,7,8,9,10,11]. As presented in [12], malnutrition among the elderly is overlooked and under-diagnosed in the United States, affecting up to 60% of hospitalized patients aged 65 and older. Additionally, Goates et al. [13] have demonstrated the financial impact malnutrition may have on several pathologies when considering the cost of medical care. Therefore, as reported by the European Society of Parenteral and Enteral Nutrition (ESPEN), systematic screening for malnutrition should be carried out using a validated tool to substantiate the diagnosis of malnutrition and as a basis for the definition of individual treatment goals and the development of a comprehensive nutritional care plan [14].

Several scientific societies as well as national and international organizations call for the early detection of malnutrition, enabling the health system to mitigate negative associated health consequences by proactively implementing corrective measures to address the patient’s nutrient deficiency. In Spain, some studies, which focused on the elderly population, show the nutritional status to which this population is exposed both living at home as well as in the institutional environment. The PREDyCES study (Prevalence of hospital malnutrition and associated costs in Spain) is an example of the observations made regarding patients’ nutritional status [15]. This ground-breaking study showed that 57% of hospitalized elderly patients ran the risk of being malnourished. Hence, the SEGG-SENPE consensus document on nutritional assessment in the elderly indicates that “malnutrition in the elderly could be partly avoided if all those maneuvers aimed at preventing the development of malnutrition or treating it early were carried out” [16].Therefore, it is imperative to consider innovative multidisciplinary approaches, such as personalized nutritional guidance systems carried out by our research group [17] for the prevention of malnutrition in hospitalized multimorbid older patients, and the use of nutritional formulas to solve malnutrition as it is a highly prevalent problem that carries significant costs for the public health system.

Currently, nutritional questionnaires are used to detect malnutrition in older adults. The most widely used for this population is the Mini Nutritional Assessment (MNA) questionnaire [8]. However, it can be time consuming for caregivers to complete this type of survey each time a patient is admitted to the hospital. Moreover, some studies have shown that women may have a higher risk of malnutrition [1,2,3,4,5], and it is necessary to have a special focus on that. Thus, identifying the factors that contribute most to malnutrition, checking the factors that may have higher impacts on older women and developing new tools, such as predictive models for assessing the nutritional status, facilitates the identification of malnourished populations and simplifies the health professionals’ work.

It is essential to consider the most suitable variables to design an efficient predictive model for the risk of malnutrition. Some studies have reported that several physiological changes associated with age, socio-economic status, and neuropsychological factors may contribute to insufficient dietary intake [18,19,20]. Regarding physiological factors, slower gastric emptying, altered hormonal responses, decreased basal metabolic rate, and altered taste and smell may also contribute to lowered energy intake [10,20,21]. In terms of socio-economic and neurophysiological aspects, other factors like marital status, social isolation, cognitive impairment, depression, and education level may be associated with malnutrition [18]. Additionally, O’Keeffe et al. [22] present a systematic review of potentially modifiable determinants of malnutrition, which involve seven domains (oral, psychosocial, medication and care, health, physical function, lifestyle, and eating). Besides, gender inequality in nutrition is also studied [1,2,3,4,5], showing that poverty, education, lack of awareness and marital status may contribute to malnutrition in women. 

However, these studies have some limitations: (i) lack of sufficient data for a representative sample, (ii) focus on a specific population with a particular disease or condition, (iii) usage of non-validated measures to determine malnutrition, (iv) ignore other potential factors, such as living conditions, or (v) focus on developing countries. 

In addition, although some studies have been carried out in the last years to develop models for the prediction of malnutrition, few studies focus on older hospitalized people, and do not focus on women cases, and the approaches centered in this population have limitations regarding their predictive capacity. For instance, in the case of the models developed by Muñoz et al. [23], a nutritional screening standard was considered for the selection of variables, which makes the predictive capacity of these models questionable.

To overcome these limitations, the current observational study aimed at identifying the key factors that contribute to malnutrition in the older adults, further studying the situation with women, and using those key factors to develop an efficient model for predicting the risk of malnutrition in this population. Furthermore, a statistical analysis was carried out to assess the effect of malnutrition on hospital admission, such as length of stay (LOS) and readmission.

## 2. Materials and Methods

### 2.1. Design and Subjects

The observational study was promoted by Asunción Klinika Hospital, located in Tolosa (north of Spain), between January 2019 and December 2019. All subjects were screened according to the following inclusion and exclusion criteria.

*Inclusion criteria:* Age > 65 years old; admitted in Asunción Klinika Hospital during 2019; and accepted the informed consent.

*Exclusion criteria:* people with morbid obesity, bulimia, and anorexia nervosa; patients whose clinical situation prevented the collection of study variables; patients on hemodialysis; and refusal to participate in the study.

### 2.2. Data Collection

Following the Joint Commission for Accreditation of Healthcare Organization’s guidelines (2011), the nutritional assessment has been performed systematically within the first 24–48 h of the patient’s admission.

The data collected to assess its effect on malnutrition were studied from two different perspectives: patient factors and hospitalization factors.

#### 2.2.1. Patient Factors

Patient factors contain three main categories: demographic information, clinical information, and laboratory information.

*Demographic information* included age (in years), sex (dichotomous variable), and potentially relevant information regarding living conditions. This last data point served to determine (i) the accessibility of their home and (ii) whether they have support at home. First, examining the home’s physical accessibility such as whether the apartment requires the use of steps or has an elevator is important since it can affect isolation [24]. Secondly, it is critical to determine whether the patients have support at home in the form of a spouse by inquiring directly about their marital status, and whether they live alone or have assistance.

*Clinical information* was recorded as follows: height (cm) was asked of the patient if they could easily recall it; otherwise, the length of the forearm was taken, and their height was estimated [25]. Current weight at the time of admission was measured with a levelled platform scale and their body mass index (BMI) was calculated as Weight/Height^2^. Other clinical information gathered included whether there was a presence of artificial nutrition in the diet, level of functional independence determined by the Barthel Index [26] ranging from 0 (fully dependent) to 100 (totally independent), and comorbidities such as diabetes, chronic obstructive pulmonary disease (COPD), congestive heart failure (CHF), multimorbidities, and pressure ulcers/sores.

*Laboratory information* included serum albumin, total cholesterol level and total lymphocyte count. Lymphocytes were measured several times during the patient’s hospital stay, however, within this study only the first documented measurement has been considered. Note that these three measurements are also used to measure CONUT malnutrition screening test [27].

#### 2.2.2. Hospitalization Factors

Additionally, hospital admission information was collected to determine firstly the effect that the type of admission (whether the admission is programmed or an emergency) may have on malnutrition (i.e., causes) and, secondly the effect that malnutrition may have on further complications (i.e., consequences).

The causalities involved the hospital unit in admission, diagnosis-related group (DRG) severity, and admission type. The hospital unit variable included ten different hospital units or departments: Anesthesia (ANES), Cardiology (CARD), Surgery (SUR), Gynecology (GINE), Internal Medicine (IM), Neurology (NEUR), Otorhinolaryngology (ORL), Traumatology (TRAU), Intensive Care Unit (ICU), and Urology (URO). Nevertheless, less than 1% of the individuals in the study were assigned to ANES, GINE, ORL, and ICU sections; thus, they were not considered for the study. DRG severity for each patient range from 1 (low severity) to 4 (high severity), and admission type was distinguished between surgical procedure and medical procedure.

The studied consequences that could be derived from malnutrition included length of stay (LOS), readmission, surgical wound dehiscence, nosocomial infection, and ulcer complications at admission. However, since less than 1% of the population from the study suffered surgical wound dehiscence, nosocomial infection, or ulcer complications at admission, only LOS and readmission were presented. LOS was calculated based on the duration of hospitalization by subtracting day of admission from day of discharge (measured in days). Readmission was calculated if the same patient was admitted within 30 days after being discharged.

### 2.3. Nutrition Screening

For the present study we performed the MNA-SF malnutrition screening test since it is validated, reliable, and intuitive to those carrying out the assessment [28,29].

This screening test is the method recommended by ESPEN [8,30]. It is the reduced version of the MNA questionnaire and consists of six questions. The questions cover mobility, stress or illness, BMI, dementia or depression, weight loss, and reduction of food intake issues. The total score of the MNA-SF ranges from 0 to 14 points (12–14 points: normal nutritional status; 8–11 points: at risk of malnutrition; 0–7 points: malnourished) [31]. 

### 2.4. Preprocessing

Data collected during the study were documented by documentarists, and consequently, missing values were limited, and few inconsistencies were found. Two height measurements, collected in cm, were incongruent (lower than 30 cm) and have not been considered. Two patients were not assessed with MNA-SF, and therefore, they have not been considered in the study. Each instance corresponds to hospital admission. Therefore, if one patient had more than one admission, the patient information could have been used in more than one instance (for each admission).

### 2.5. Statistical Analysis

To identify the effect of each factor on malnutrition, several bivariate analyses were carried out by exploring the relationships between the MNA-SF variable and each of the other attributes in the dataset. For the bivariate analyses, in the case of a categorical variable with a numerical one, the one-way analysis of variance (ANOVA) was used to determine whether there were any statistically significant differences between the means of the analyzed variables. In the case of two numerical variables, correlations were calculated using the Pearson coefficient. Results of the study were considered statistically significant when *p* < 0.001. The ANOVA method performs better if the distribution of the numerical variable follows a normal distribution, which is not the case in this study (Shapiro-Wilk test [32] result with a *p*-value < 0.001 and W = 0.925). Therefore, we normalized the principal variable (MNA-SF) for the analysis.

### 2.6. Predictive Model Design

In order to design an efficient predictive model, first, a recursive feature elimination (RFE) algorithm was used to obtain an optimum number of variables. For the design of this algorithm, a random forest model and a repeated cross-validation resampling method were used. After this selection, various machine learning (ML) algorithms were used to develop the different risk predictive models. In this process, 80% of the data (798 patients’ data) were used for training the models, and 20% (200 patients’ data) for the testing. To select the final models to be validated in a future study, the main indicators (sensitivity, specificity, and ROC-AUC) of all the obtained models were compared.

## 3. Results

Of the 1000 patients included in the study, two were not evaluated with MNA-SF, so the following results consider 998 patients’ information, of which 505 were males and 493 were females with a mean age of 81 years old (males 79.87 ± 7.7; females 82.24 ± 7.9). The following Section 3.1 presents the results of the main features that have an impact on malnutrition and Section 3.2 describes the results of the developed predictive models.

### 3.1. Patient Factors

As it is presented in Table 1, the impact of eighteen demographic, clinical, and laboratory test features have been studied. The shown statistics are presented for the total sample, but also for each of the MNA-SF categories, i.e., malnutrition, risk of malnutrition, and normal. Considering all patients, 443 (44.4%) were well nourished (normal), 417 (41.8%) were detected with risk for malnutrition, and 138 (13.8%) patients were considered malnourished.

Table 1 shows that female gender is associated with higher risk of malnutrition. Besides, age, marital status, assistance, and BARTHEL, also show an effect on malnutrition.

### 3.2. Women Factors

Further study has been carried out to determine which factors contribute to malnutrition on older women (Table 2), since the number of women malnourished is significantly higher than men (Table 1). In contrast with Table 1, in Table 2 age and marital status have not such an impact for women (*p* > 0.001). This may be caused by the implicit relation in the dataset between sex factor and age and marital status, since the mean age of women in the dataset is elder, and this is also related to the marital status with higher number of widowed women than men (58% and 18.6%, respectively). However, other factors, such as the *assisted*, *BMI*, *Barthel* and *serum albumin* do present *p* < 0.001.

### 3.3. Hospitalization Factors

As presented in Section 2.2, in addition to patient factors, information obtained when patients are admitted to hospital has been analyzed. In Table 3 the relationship between the potential admission-related causalities and malnutrition are represented, showing a high association in the three cases.

Finally, Table 4 presents the potential consequences of malnutrition, showing the *p*-value for LOS and readmission clinical outcomes. Malnourished patients demonstrated a significantly higher LOS rate (9.76 ± 8.34 days), compared to the LOS for patients with risk of malnutrition (7.77 ± 6.43 days) and with well-nourished patients (6.46 ± 5.53 days), while it does not seem to have a significant effect on readmission rates.

### 3.4. Malnutrition Risk Predictive Model

The accuracy results for the RFE analysis ranged from 55% to 67%, as shown in Figure 1a. Selecting the most accurate result, a total number of sixteen variables were used for the development of the models. The importance of each variable in the trained classification model has been analyzed and ordered according to it in Figure 1b, being height, age and weight at admission the three most relevant variables. Among these sixteen variables, two attributes not mentioned in Section 3.1, Section 3.2 and Section 3.3 can be found. If elevator floor combines the floor and elevator variables, converting the value in floor variable to 0 if the patient has an elevator. Days from previous admission refers to the number of days passed from the date of the previous admission to the date of the current admission, set to 0 when the patient is admitted for the first time.

Being three the possible values of the predicting parameter MNA (malnutrition, risk of malnutrition, and normal), two final models were developed: (i) a model designed for the prediction of no risk vs risk or high risk of malnutrition (as a group), and (ii) a second model, in the case of obtaining the second option in the first model, for the prediction of risk vs high risk of malnutrition (see Figure 2). To create the models, seven different machine learning algorithms were implemented and compared. To assess the performance of these models, two validation approaches have been used, internal 10-fold cross-validation and external validation using the test set (20% of initial dataset, N = 198); their sensitivity, specificity, and ROC-AUC values are shown in Table 5 and Table 6. Note that the model’s outcome is a percentage, and this must be translated to a value. For that, the threshold (THR) determined for Model 1 is THR = 0.5 and for Model 2 THR = 0.3, which obtained the best results from a medical point of view. Random forest and gradient boosting algorithms were compared for developing Model 2 (Table 6), as they showed much better results than the other algorithms in Model 1 (Table 5).

Finally, the two models with the highest AUC values were selected: the random forest model for the first prediction and the gradient boosting model for the second one with AUC values of 0.758 and 0.735 respectively.

After combining the two models according to the scheme in Figure 2, we analyzed how the final model stratified each patient (i.e., Predicted) vs. reality (i.e., Real) using the test set (N = 198), as shown in Table 7 confusion matrix. The diagonal in *green* shows the number of patients the model has stratified correctly. The two contiguous diagonals in *yellow* present when the model failed with a “low” error (E1), i.e., normal vs. risk; risk vs. malnourished. Finally, the *orange* positions show “high” errors (E2), i.e., normal vs. malnourished. As a result, we have an accuracy of 52% (percentage of patients who are in the green diagonal). This value may seem discrete, but on the other hand, the prevalence of the E2 error, which is the one to be avoided, is 5%.

## 4. Discussion

In the present population-based cohort study of older adults aged ≥65, 44.4% of the studied population is well nourished, whereas 41.8% is at risk of malnutrition and 13.8% has malnutrition based on the MNA-SF screening test. This picture comes out worse when it comes to women with 36.1% of women well nourished, 47.5% at risk and 16.4% with malnutrition, as also presented in [1].

In this observational study we examined the association between risk of malnutrition and a wide range of variables including demographic, clinical (chronic diseases) and laboratory data. From the demographic point of view, our findings suggested that older age, female gender, having assistance at home, and being dependent (via Barthel index) are all factors associated with malnutrition. However, it is important to note that the MNA-SF screening test applied in our study makes use of BMI, and asks for mobility, which is related with BARTHEL and assisted living factors of our study. Therefore, we cannot say that BMI is a key indicator of malnutrition, and BARTHEL and assisted living factors may also be influenced by the MNA-SF mobility question.

Regarding architectural barriers, although the absence of elevator does not have *p* < 0.001, is close to it with a value of *p* = 0.007, and hence, we should consider it since it may contribute patients to be more isolated [23], and hence, with less access to food. 

Note that from the clinical point of view, some of the studied patient factors are causative risk factors, such as age, sex and assisted, and others are consequences of malnutrition, such as BMI value. However, other factors may not be clear, such as the dependency situation (i.e., BARTHEL index), since they could be both causative and consequence factors of malnutrition or the case of albumin, as both (albumin level and malnutrition) are related to inflammation which can describe the association but not the causality between them as described by Evans et al. [33].

Considering the threshold of *p* < 0.001, only patients with pressure ulcers seem to have a higher risk of malnutrition, but multimorbidity shows a *p* = 0.003, which is close to the value, aligned with the literature [34]. However, diabetes, heart failure (HF), or chronic obstructive pulmonary disease (COPD) single diseases do not seem to be associated with malnutrition. In addition, upon examining obtained laboratory data, only serum albumin presents a high correlation with malnutrition. 

Regarding hospitalization factors, the present study demonstrates the high impact of the type of admission on malnutrition. The hospital unit factor, which is highly related with the patient diagnosis, shows that patients admitted in cardiology, neurology, traumatology, and urology have a lower prevalence of malnourished patients, while in internal medicine units, the number is higher. This way, internal medicine seems to be a unit with higher risk of malnourishment. DRG severity is another factor that is highly related to malnutrition, where more severe patients show a higher risk of malnutrition. Besides, medical admissions are also highly correlated with malnutrition, since they are more related to medical conditions that affect the patient as a whole, while surgical admissions are inversely related, since they used to be programmed admissions. 

To the best of our knowledge, this is the first study in the Basque Region to evaluate the risk of malnutrition in the elderly population when admitted to the hospital. Other studies in Spain examining the risk of malnutrition [30,32] suggest that undernutrition rates are over 40% using the CONUT and SGA screening tests. On the other hand, other Spanish studies performed using the MNA screening test obtain similar results to ours, with a low prevalence of malnutrition (around 7% in the study population), whereas the risk of malnutrition reaches nearly half of the study population (49%) [35,36,37]. 

As expected, the Barthel index, which determines the dependency level of a person, is associated with malnutrition. However, chronic diseases do not seem to influence the risk of malnutrition in the elderly population. In addition, increased age correlates with malnutrition. This could also be related to cognitive functioning, as suggested by Katsas et al. [19], since increased age may cause a decline in cognitive functioning, particularly in memory. Memory is associated with many aspects of daily life, such as eating, and consequently can lead to an increased risk of malnutrition [38,39]. 

Aligned with other studies, we found that women had a higher risk of malnutrition than men [2,3,4,5]. Therefore, further analysis was performed to identify the main differences that may exist between both groups (see Table 2 and Table A1 in Appendix A). We noted that marital status in Table 1 seems to be a relevant factor. However, when analyzing men and women separately, it appears that marital status does not have such a great impact. The reason behind this is that the number of widowed women is much higher than the number of widowed men (i.e., there is a high correlation between gender and marital status, *p* < 0.001). On the other hand, age seem to have a higher impact on men (*p* < 0.001) than women, although for women it is close to *p* < 0.001 (*p* = 0.002). For this reason, we conclude that age and sex are the primary factors that contribute to malnutrition, but not marital status in and of itself. In addition, we analyzed the distribution of the MNA variable for both sex factors (woman and man), finding worse nutritional conditions for the female gender. This higher prevalence of malnutrition risk among women may be related to social issues related to social and financial living as suggested in [4], or depressive states being more common in women as found in [2,3].

Additionally, the results show that malnutrition is associated with longer hospital LOS, aligned with [10,11,40]. This extended LOS leads to a significant increase in hospital costs [11]. However, there is a high correlation between LOS and DRG severity, and hence, the association between LOS and malnutrition is not clearly demonstrated.

Regarding readmissions, in contrast with [10], this study does not present significantly higher readmission rates in patients with malnutrition or those who are at risk of malnutrition, compared to those who are well-nourished.

Finally, identifying all these risk factors has led to the development of a model to predict the risk of malnutrition in hospitalized older adults. Agreeing with [41], fewer features can allow machine learning algorithms to run more efficiently (less space or time complexity) and be more effective, as some algorithms can be misled by irrelevant input features, resulting in worse predictive performance. Thus, after the application of a recursive feature elimination (RFE) algorithm, sixteen variables were selected for the development of the models. After comparing the ROC-AUC values of all the designed models, the random forest and gradient boosting algorithms obtained the highest accuracy values. However, concurring with [42], discrimination is assumed to be useful if AUC ≥ 0.75, and hence the obtained models present fair predictive ability. This assumption can be confirmed by the interpretation of the confusion matrix, which shows that the model is able to predict correctly with an accuracy of 51.6%. Although this result is not as expected, we have almost no extreme errors and, considering that it is a three-way classification, it may be suitable for its use.

Among the limitations of this study is the lack of availability of each separate MNA-SF response to examine answers independently. Information regarding cognitive functioning, education level, and socioeconomic status has not been collected, which could also be of interest to the current study. Throughout the study, information regarding complications, such as surgical wound dehiscence, nosocomial infection, and ulcer complications upon admission was collected to determine the impact of malnutrition on patients. Nevertheless, since the number of patients that suffered these complications was too small, no conclusions could be drawn, and consequently, these results have not been presented in the study.

Regarding the strengths of the study, we used a representative sample of older adults admitted to the hospital, with almost 50% of them women, in contrast with other studies which used specific subgroups of older people. Additionally, the results show a clear correlation of the factors with a higher impact on malnutrition, and we also examined the primary reason why, since the sample data may influence the results.

The findings of the present research emphasize the urgent need for physicians and clinical institutions to be aware of the high prevalence of malnutrition in elderly patients, with a greater focus on women, and the influence of the admission information in malnutrition. Therefore, health experts should perform nutritional screening for all older adults as part of secondary prevention, with a special focus on more vulnerable populations, such as widowed elderly women who require assistance at home, those who are considered dependent as categorized by the Barthel index, or patients admitted in internal medicine with high DRG severity. To those who are more vulnerable, nutrition counseling and support should be offered.

## 5. Conclusions

The main risk factors identified for the prevalence of malnutrition are female gender, being assisted at home, having high scores of dependency, no access to an elevator, and presenting pressure ulcers and multimorbidity. In addition, patients with malnutrition showed a significant increase in hospital LOS with respect to patients who are well nourished, although it is also related with the admission DRG severity. Although the developed malnutrition risk predictive models present fair accuracy results, further research needs to be conducted to validate them clinically.

## Figures and Tables

**Figure 1 geriatrics-07-00105-f001:**
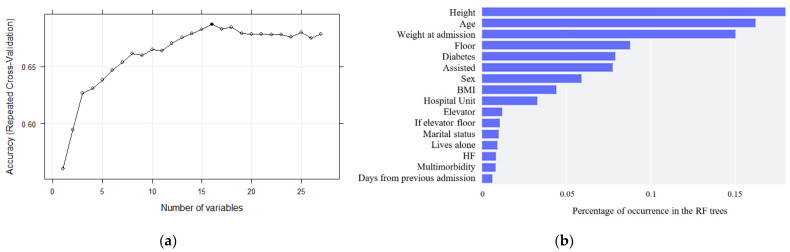
(**a**) Accuracy of the different amounts of variables by repeated cross-validation for RFE; and (**b**) feature list sorted by relevance.

**Figure 2 geriatrics-07-00105-f002:**
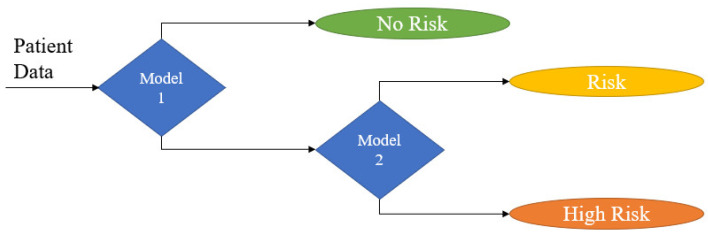
Diagram of the logic used for the prediction of risk of malnutrition.

**Table 1 geriatrics-07-00105-t001:** Relationship between demographic, clinical, and laboratory tests, and malnutrition as evaluated by MNA-SF (N = 998). In this table, descriptive statistics are presented as percentages for categorical variables, and mean values and standard deviations for numerical variables. Attributes were considered statistically significant when *p*-value < 0.001. Acronyms on the table: MNA, Mini Nutritional Assessment; COPD, chronic obstructive pulmonary disease.

Variable		Total Sample	MNA (Malnutr.)	MNA (Risk)	MNA (Normal)	*p*-Value
Percentage		100	13.8	41.8	44.4	
Age (years)		81 ± 7.9	84 ± 7.8	81.9 ± 7.6	79.3 ± 7.9	<0.001
Sex (% men)		50.6	41.3	43.9	59.8	<0.001
Elevator (% yes)		63.7	71	63.6	61.6	0.007
Floor (numeric)		2.14 ± 1.8	2.22 ± 1.8	2.22 ±1.9	2.03 ±1.7	0.296
Marital Status	Married (%)	47.7	34.8	43.4	55.8	<0.001
	Single (%)	14.2	14.5	15.8	12.6	
	Widow (%)	38.1	50.7	40.8	31.6	
Assisted (%)		54.3	77.5	59.5	42.2	<0.001
Lives Alone (%)		12.5	5.8	15.1	12.2	0.113
Body Mass Index (kg/m^2^)		26.9 ± 5.3	23.4 ± 4.6	26.2 ± 4.9	28.7 ± 5.1	<0.001
BARTHEL		75.1 ± 27.4	48.7 ± 34.1	72.5 ± 26.0	85.8 ± 18.8	<0.001
Diabetes (% yes)		28	33.3	27.6	26.6	0.134
Heart Failure (% yes)		23.2	27.5	25.4	19.9	0.047
COPD (% yes)		14.7	13.8	15.6	14.6	0.804
Multimorbidity (% yes)		10.9	18.1	12.2	7.45	0.003
Pressure Ulcers (% yes)		6.1	13.8	5.51	4.3	<0.001
Serum Albumin (g/dL)		3.07 ± 0.5	2.85 ± 0.5	3.08 ± 0.5	3.16 ± 0.5	<0.001
Cholesterol (mg/dL)		153 ± 45.1	153 ± 60.3	153 ± 40	152 ± 43.4	0.578
Total Lymphocytes (/mL)		1.53 ± 1.1	1.64 ± 1.9	1.58 ± 1	1.45 ± 0.8	0.257

**Table 2 geriatrics-07-00105-t002:** Relationship between demographic, clinical and laboratory tests, and malnutrition as evaluated by MNA-SF for women (N = 493). Attributes were considered statistically significant when *p*-value < 0.001. Acronyms on the table: MNA, Mini Nutritional Assessment; COPD, chronic obstructive pulmonary disease.

Variable		Total Sample	MNA (Malnutr.)	MNA (Risk)	MNA (Normal)	*p*-Value
Percentage		100	16.4	47.5	36.1	
Age (years)		82.2 ± 7.9	84.7 ± 7.5	82.7 ± 7.9	80.6 ± 7.9	0.002
Elevator (% yes)		65.5	72.8	65.4	62.3	0.086
Floor (numeric)		2.12 ± 1.8	2.26 ± 1.7	2.23 ±2	1.93 ±1.6	0.187
Marital Status	Married (%)	30.2	21	29.1	36	0.059
	Single (%)	11.8	12.3	12.4	10.7	
	Widow (%)	58	66.7	58.5	53.4	
Assisted (%)		62.1	77.8	65.8	50	<0.001
Lives Alone (%)		14.6	4.9	15.8	17.4	0.038
Body Mass Index (kg/m^2^)		264 ± 5.6	23.7 ± 4.5	25.9 ± 5.5	28.1 ± 5.7	<0.001
BARTHEL		70.9 ± 28.5	46.5 ± 34.5	70.2 ± 26.4	82.9 ± 19.8	<0.001
Diabetes (% yes)		24.7	32.1	22.6	24.1	0.058
Heart Failure (% yes)		19.3	27.2	19.7	15.2	0.044
COPD (% yes)		7.5	8.6	6	9	0.576
Multimorbidity (% yes)		11.8	18.5	11.1	9.6	0.015
Pressure Ulcers (% yes)		7.1	13.6	6.4	5.1	0.009
Serum Albumin (g/dL)		3.05 ± 0.5	2.81 ± 0.5	3.09 ± 0.5	3.14 ± 0.5	<0.001
Cholesterol (mg/dL)		162 ± 40.4	149 ± 34.7	164 ± 40.1	166 ± 43.2	0.027
Total Lymphocytes (/mL)		1.72 ± 1.4	1.81 ± 2.3	1.72 ± 1.2	1.68 ± 0.9	0.932

**Table 3 geriatrics-07-00105-t003:** Relationship between hospital admission information and malnutrition as evaluated by MNA-SF (N = 998). Attributes were considered statistically significant when *p*-value < 0.001. Acronyms: MNA, Mini Nutritional Assessment; DRG, Diagnosis-Related Group.

Variable		Total Sample	MNA (Malnutr.)	MNA (Risk)	MNA (Normal)	*p*-Value
Hospital Unit	Cardiology (%)	6.31	4.35	6.23	7	<0.001
	Surgery (%)	13.7	10.9	14.9	13.5	
	Internal Medicine (%)	56.9	71	60.7	49	
	Neurology (%)	7.61	5.8	6.23	9.48	
	Traumatology (%)	11.2	7.25	8.15	15.4	
	Urology (%)	3.51	0.72	3.12	4.74	
DRG Severity	1 (%)	27.6	22.5	23.7	32.7	<0.001
	2 (%)	48.7	44.2	53	46	
	3 (%)	21.8	29	21.1	20.3	
	4 (%)	1.9	4.35	2.16	0.9	
Admission Type	Medical (%)	91.6	94.2	95	87.6	<0.001
	Surgical (%)	8.42	5.8	5.04	12.4	

**Table 4 geriatrics-07-00105-t004:** Associations of malnutrition evaluated by MNA-SF with Length of Stay and readmission (N = 998). Attributes were considered statistically significant when *p*-value < 0.001. Acronyms on the table: MNA, Mini Nutritional Assessment; LOS, Length of Stay.

Variable	Total Sample	MNA Maln.	MNA Risk	MNA Normal	*p*-Value
LOS	7.46 ± 6.45	9.76 ± 8.34	7.77± 6.43	6.46 ± 5.53	<0.001
Readmission (% yes)	9.7	9.4	9.8	9.7	0.932

**Table 5 geriatrics-07-00105-t005:** Model 1 prediction results (no risk vs risk or high risk). Acronyms on the table: AUC, Area Under the ROC Curve; ML, Machine Learning; SVM, Support Vector Machine.

ML Algorithm	Sensitivity	Specificity	AUC
Random Forest	62.9%	74.2%	**0.758**
K-Nearest Neighbor	53.3%	63.8%	0.639
Logistic Regression	49.5%	62.4%	0.559
Gradient Boosting	60.5%	70.8%	**0.733**
Linear SVM	45.9%	78.3%	0.682
C5.0	57.9%	66.1%	0.644
Neural Network	26.0%	89.9%	0.682

**Table 6 geriatrics-07-00105-t006:** Model 2 prediction results (risk vs high risk). Acronym: AUC, Area Under the ROC Curve.

ML Algorithm	Sensitivity	Specificity	AUC
Random Forest	19.1%	97.1%	0.734
Gradient Boosting	20.0%	97.0%	0.735

**Table 7 geriatrics-07-00105-t007:** Confusion matrix where it is compared the classifier prediction with the reality in the test set. It is coloured in the following way: in *green* are those that are correct, in *yellow,* are wrong but with low error (E1,) and in *orange* are wrong with high error (E2).

		Real		
		Malnutrition	Risk	Normal
Predicted	Malnutrition	16	16	6
	Risk	11	39	34
	Normal	4	24	48

## Data Availability

Not applicable.

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
