# Peer review of "Key Factors and AI-Based Risk Prediction of Malnutrition in Hospitalized Older Women"

_geriatrics, 2022, doi:10.3390/geriatrics7050105_

Round 1
Reviewer 1 Report
The presented in the said manuscript approach to the topic of malnutrition among older Spanish women seems to be innovative and interesting. However, although the authors both in the abstract and in discussion draw the attention to the impact of the female gender, they did not make an effort to present any hypothesis concerning this phenomenon (e.g., more lifelong care concerning the body image in women as compared men). In this context, also the statement placed in the Introduction "What is more, the mere fact of being women is one of the risk factors for developing these habits" should be developed.
Abstract
"The aim of this study was to evaluate the prevalence of malnutrition in older adults, specially focusing on older women, identify its key factors and develop a malnutrition risk predictive model. The observation was held over one year examining 1,000 patients (493 women)".
This sentence should be more precise; the note about 1,000 pts is unnecessary; e.g. " The study group consist of 493 older hospitalized women".
Material
Were there, in the examined patients' comorbidities, that could have impact on the malnutrition occurrence, any chronic kidney disease or neoplastic cases?
Discussion
Line 313: "our findings suggested that being elderly, being a woman, having assistance at home, low BMI and being dependent (via Barthel index) are all factors associated with malnutrition" - BMI is an obvious result of malnutriton, not a reason. (see line 325). The expressions "being elderly, being a woman" are akward. Please, replace them in all text by "older age, female gender".
"Information regarding cognitive functioning, education level, and socioeconomic status has not been collected, which could also be of interest to the current study." That is true and I am curious why these, very easy to be collected data, was omitted?
References
The predominating number of the quoted papers were published prior 2020 year. Please, supply the reference list with the more current publications.
Author Response
Dear reviewer,
The authors are very indebted to the Reviewers, for their detailed, inspiring, and constructive criticism. In the revised version, we have addressed all the comments by Reviewers, thus achieving a substantially improved version of our paper.
In the following, we briefly summarize the main improvements of the first version. Then, we will address one-by-one all the points risen by Reviewers (in blue).
MAIN CHANGES
On the basis of the Reviewers’ comments, we have improved our original submission along several main directions.
- We have shortened the paper
- Making the paper clearer and more readable
- Removing not necessary, and in some way “distracting” figures, since the Figures did not show any relevant results
- Removing “Artificial Diet” variable from table, since it did not have enough data
- The main goals and contributions of our work are better clarified and motivated. In particular,
- We have emphasized the women nutritional status in the abstract
- We have rephrased some sentences in order to clarify the text
One-by-one analysis Reviewers’ comments
Black text: reviewer’s comment
Blue text: authors’ response
- "The aim of this study was to evaluate the prevalence of malnutrition in older adults, specially focusing on older women, identify its key factors and develop a malnutrition risk predictive model. The observation was held over one year examining 1,000 patients (493 women)".
This sentence should be more precise; the note about 1,000 pts is unnecessary; e.g. "The study group consist of 493 older hospitalized women".
Thank you for the comment. We agree that, being a paper centred in women, we should be more concrete in describing the subjects in the abstract. Thus, we have changed this sentence for “The study group consist of 493 older women admitted to the Asunción Klinika Hospital in the Basque Region (Spain)” in order to be more precise.
- Were there, in the examined patients' comorbidities that could have impact on the malnutrition occurrence, any chronic kidney disease or neoplastic cases?
This information would be very interesting, but we only have information about the hospital unit (or department) where the patients are referred in the admission process.
- Line 313: "our findings suggested that being elderly, being a woman, having assistance at home, low BMI and being dependent (via Barthel index) are all factors associated with malnutrition" - BMI is an obvious result of malnutrition, not a reason. (see line 325). The expressions "being elderly, being a woman" are awkward. Please, replace them in all text by "older age, female gender".
You are right. We have changed those awkward expressions by the suggested ones in all text (lines 30, 220, 314, and 420), and we also have removed BMI from 313 and 421 lines.
- "Information regarding cognitive functioning, education level, and socioeconomic status has not been collected, which could also be of interest to the current study." That is true and I am curious why these, very easy to be collected data, was omitted?
This information was not considered by the medical experts during the study design.
- The predominating number of the quoted papers were published prior 2020 year. Please, supply the reference list with the more current publications.
Thank you for your comment. We have done our best to include more updated references. However, some references are the basis of some principles, and others were quite complex to find a replacement after 2020. Therefore, we have modified few references.
Thank you again for your time.
Best Regards
Reviewer 2 Report
This manuscript presents an elegant observational study of a Basque Region population aimed to detect the prevalence of malnutrition in older patients amitted to hospital. Several variables were considered, statistical analysis was correctly used and conclusions well argued, although non completely justifiable by a clinical point of view. To be an old dependent woman is a "certeinty" to be at risk of malnutrition
Author Response
Dear reviewer,
The authors are very indebted to the Reviewers, for their detailed, inspiring, and constructive criticism. In the revised version, we have addressed all the comments by Reviewers, thus achieving a substantially improved version of our paper.
In the following, we briefly summarize the main improvements of the first version.
MAIN CHANGES
On the basis of the Reviewers’ comments, we have improved our original submission along several main directions.
- We have shortened the paper
- Making the paper clearer and more readable
- Removing not necessary, and in some way “distracting” figures, since the Figures did not show any relevant results
- Removing “Artificial Diet” variable from table, since it did not have enough data
- The main goals and contributions of our work are better clarified and motivated. In particular,
- We have emphasized the women nutritional status in the abstract
- We have rephrased some sentences in order to clarify the text
Please, kindly let us know whether the response is adequate, since we are happy to follow any further instructions you may want to give us.
Thank you again for your time.
Best Regards
Reviewer 3 Report
There are some minors corrections
In lines 2-3 your article title
"Key Factors and AI-based Risk Prediction of Malnutrition in Hospitalized Older Women."
However, lines (224–226) of your article, you compare both genders. Can you explain it?
In line 107
(Inclusion criteria: Age > 65 years old), if there is some standard formula for including above 65-year old women, why not you include age <65 ?
In table 2. If there is no sample of Artificial Diet (% yes) MNA-SF for women is collected, and then it is necessary to add them, why?
However variable with Artificial Diet (% yes) MNA-SF for Men is 0.4% in Table A1.
In line (114-115)
“The collection of data and nutritional assessment has been performed systematically within the first 24-48 hours of the patient's admission”
Is two days maximum length of stay? Enough for calculating relative risk of Malnutrition patients.
In line 267
Graph (b) not representation of variable along x-axis. Define proper name of variables on both graph (a) and (b).
Author Response
Dear reviewer,
The authors are very indebted to the Reviewers, for their detailed, inspiring, and constructive criticism. In the revised version, we have addressed all the comments by Reviewers, thus achieving a substantially improved version of our paper.
In the following, we briefly summarize the main improvements of the first version. Then, we will address one-by-one all the points risen by Reviewers (in blue).
MAIN CHANGES
On the basis of the Reviewers’ comments, we have improved our original submission along several main directions.
- We have shortened the paper
- Making the paper clearer and more readable
- Removing not necessary, and in some way “distracting” figures, since the Figures did not show any relevant results
- Removing “Artificial Diet” variable from table, since it did not have enough data
- The main goals and contributions of our work are better clarified and motivated. In particular,
- We have emphasized the women nutritional status in the abstract
- We have rephrased some sentences in order to clarify the text
One-by-one analysis Reviewers’ comments
Black text: reviewer’s comment
Blue text: authors’ response
Please, kindly let us know whether the response is adequate, since we are happy to follow any further instructions you may want to give us.
- In lines 2-3 your article title "Key Factors and AI-based Risk Prediction of Malnutrition in Hospitalized Older Women." However, lines (224–226) of your article, you compare both genders. Can you explain it?
Thank you for your comment. We have considered your comment carefully, and we have decided to remove that part since it does not contribute much to the study results.
- In line 107 (Inclusion criteria: Age > 65 years old), if there is some standard formula for including above 65-year old women, why not you include age <65 ?
Describing an older adult as “elderly” should be based on physical health and medications rather than chronological age. There is no standard definition for the term elderly. However, the inclusion criteria for the study needed to define a threshold and based on other studies, medical experts decided to determine this age as a threshold.
- In table 2. If there is no sample of Artificial Diet (% yes) MNA-SF for women is collected, and then it is necessary to add them, why? However variable with Artificial Diet (% yes) MNA-SF for Men is 0.4% in Table A1.
Thank you so much for this review. It was a variable that was collected, but as noted by the reviewer, it is not a relevant variable since the number of population with it is really small.
- In line (114-115) “The collection of data and nutritional assessment has been performed systematically within the first 24-48 hours of the patient's admission”. Is two days maximum length of stay? Enough for calculating relative risk of Malnutrition patients.
Thank you for your comment. Maybe the sentence it is not clear enough, but we want to express that the nutritional screening is performed in the first two days of patient admission as recommended by different guidelines (as the one mentioned in the sentence). To clarify the sentence, we have change it with: “Following the Joint Commission for Accreditation of Healthcare Organization´s guidelines (2011), the nutritional assessment has been performed systematically within the first 24-48 hours of the patient's admission.”
In order to reinforce this, American Society for Parenteral and Enteral Nutrition has defined nutritional screening as “A process to identify an individual who is malnourished or who is at risk for malnutrition to determine if a detailed nutrition assessment is indicated”. Screening is best done within 24 to 48 hours of hospital admission by the treating doctors. Besides, international guidelines recommend that every patient should be screened for malnutrition within 24–48 hours of ICU admission.
- In line 267 Graph (b) not representation of variable along x-axis. Define proper name of variables on both graph (a) and (b).
Thank you for your comment. Sorry for not defining x-axis titles properly in this Figure 2. We have changed the (a) graph x-axis title to “Number of variables” and included the (b) graph x-axis title “Percentage of occurrence in the RF trees” described in the figure caption to the graph (and removed from the caption).
Thank you again for your time.
Best Regards
Reviewer 4 Report
Very interesting manuscript analyzing the most widespread and underestimated geriatric syndrome .
I agree with the call to disseminate in the clinical evaluation of the geriatric patient also the nutritional status,which is not time-consuming and economical.
Surely studies,in time,will be more focused in the evaluation of some important functional aspects. Albuminemia is a faithful mirror of malnutrition ,and easy to measure,however,it may not be the only one.
It will also be important in the future to assess the status of muscle mass ,by simple hand grip.
This would add more broadly pathophysiologic significance to the patient's nutritional status.
Author Response
Dear reviewer,
The authors are very indebted to the Reviewers, for their detailed, inspiring, and constructive criticism. In the revised version, we have addressed all the comments by Reviewers, thus achieving a substantially improved version of our paper.
In the following, we briefly summarize the main improvements of the first version. Then, we will address one-by-one all the points risen by Reviewers (in blue).
MAIN CHANGES
On the basis of the Reviewers’ comments, we have improved our original submission along several main directions.
- We have shortened the paper
- Making the paper clearer and more readable
- Removing not necessary, and in some way “distracting” figures, since the Figures did not show any relevant results
- Removing “Artificial Diet” variable from table, since it did not have enough data
- The main goals and contributions of our work are better clarified and motivated. In particular,
- We have emphasized the women nutritional status in the abstract
- We have rephrased some sentences in order to clarify the text
One-by-one analysis Reviewers’ comments
Black text: reviewer’s comment
Blue text: authors’ response
Please, kindly let us know whether the response is adequate, since we are happy to follow any further instructions you may want to give us.
Very interesting manuscript analyzing the most widespread and underestimated geriatric syndrome.
I agree with the call to disseminate in the clinical evaluation of the geriatric patient also the nutritional status, which is not time-consuming and economical.
Surely studies,in time,will be more focused in the evaluation of some important functional aspects. Albuminemia is a faithful mirror of malnutrition ,and easy to measure,however,it may not be the only one.
It will also be important in the future to assess the status of muscle mass ,by simple hand grip.
This would add more broadly pathophysiologic significance to the patient's nutritional status.
Thank you for all your comments. We will take into consideration these comments for future analysis in order to have further results.
Thank you again for your time.
Best Regards